# Effects of *Bacillus subtilis* on Cucumber Seedling Growth and Photosynthetic System under Different Potassium Ion Levels

**DOI:** 10.3390/biology13050348

**Published:** 2024-05-15

**Authors:** Chun Li, Qingpan Zeng, Yuzhu Han, Xiaofu Zhou, Hongwei Xu

**Affiliations:** 1Jilin Provincial Key Laboratory of Plant Resource Science and Green Production, Jilin Normal University, Siping 136000, China; 15590267115@163.com (C.L.); 13474205037@163.com (Q.Z.); 2School of Environment and Resources, Biotechnology, Dalian Minzu University, Dalian 116620, China; hanyuzhu6969@126.com

**Keywords:** *B. subtilis*, potassium, chlorophyll fluorescence parameters, photosynthetic characteristics, cucumber

## Abstract

**Simple Summary:**

The experiment was conducted to investigate the effects of *Bacillus subtilis* on the growth and photosynthetic system of cucumber seedlings under different potassium levels. The pot experiment was conducted with “Xinjin 4” as the test material, and the two-factor experiment was designed. The two factors were different concentrations of potassium ion and *Bacillus subtilis* treatment. The effects of different treatments on growth, photosynthetic characteristics, root morphology and chlorophyll fluorescence parameters of cucumber seedlings were studied. The results showed that *Bacillus subtilis* had the most significant effect on cucumber seedling growth and leaf photosynthesis when the concentration of potassium ion was 0.2 g/pot. This study provided a theoretical basis for further utilizing *Bacillus subtilis* to make microbial fertilizer and improve the nutrient absorption efficiency of cucumber to promote the development of agriculture.

**Abstract:**

Potassium deficiency is one of the important factors restricting cucumber growth and development. This experiment mainly explored the effect of *Bacillus subtilis* (*B. subtilis*) on cucumber seedling growth and the photosynthetic system under different potassium levels, and the rhizosphere bacteria (PGPR) that promote plant growth were used to solubilize potassium in soil, providing theoretical support for a further investigation of the effect of biological bacteria fertilizer on cucumber growth and potassium absorption. “Xinjin No. 4” was used as the test material for the pot experiment, and a two-factor experiment was designed. The first factor was potassium application treatment, and the second factor was bacterial application treatment. The effects of different treatments on cucumber seedling growth, photosynthetic characteristics, root morphology, and chlorophyll fluorescence parameters were studied. The results showed that potassium and *B. subtilis* had obvious promotion effects on the cucumber seedling growth and the photosynthesis of leaves. Compared with the blank control, the *B. subtilis* treatment had obvious effects on the cucumber seedling height, stem diameter, leaf area, total root length, total root surface area, total root volume, branch number, crossing number, g_s_, WUE, Ci, and A; the dry weight of the shoot and root increased significantly (*p* ≤ 0.05). Potassium application could significantly promote cucumber growth, and the effect of *B. subtilis* and potassium application was greater than that of potassium application alone, and the best effect was when 0.2 g/pot and *B. subtilis* were applied. In conclusion, potassium combined with *B. subtilis* could enhance the photosynthesis of cucumber leaves and promote the growth of cucumber.

## 1. Introduction

Cucumber, *Cucumis sativus* L., is an annual herb of Cucurbitaceae [1], which has important economic and biological value and is one of the top ten cultivated vegetable crops in the world. Cucumbers are popular and cultivated worldwide [2,3] for their delicious taste, high water content, and richness in a variety of nutrients [4]. Cucumber is one of the main crops in facility agriculture; because of its short growth cycle and good economic benefits, it has received much attention and has been widely planted. Through the application of microbial fertilizer, efficient, stable, and sustainable production can be achieved, making important contributions to agricultural development and market supply. Cucumbers are considered a “potassium-loving” crop that requires a high proportion of fertilizers for good growth and high yield [5,6]

Potassium is an element that is necessary for plant growth and metabolism [7,8]. K^+^ is essential for crop ripening, especially in fruit development [9]. Potassium ion is involved in many physiological and biochemical processes of plant growth and development, including osmotic regulation, stomatal movement, material transport, enzyme activation, protein synthesis, and ionization balance, and plays an important role in all aspects of plant growth and development [8,10,11]. According to the availability of potassium to plants, it can be divided into three major forms: mineral potassium, slow-acting potassium, and fast-acting potassium. Slow-available potassium is equivalent to non-exchangeable potassium, and fast-available potassium is equivalent to exchangeable potassium and water-soluble potassium. Slow potassium and mineral potassium have a high accumulation in the soil, but these two elements cannot be directly absorbed by plants; fast potassium mainly exists in the form of ions, and can be directly absorbed and converted by plants. In soil, K_2_O can be used as potassium fertilizer to provide potassium for crops but the content of fast potassium in the soil is only 0.07–0.35%, far from meeting the needs of crops for potassium [12,13,14]. Therefore, in order to maximize the sustainable production of the soil, farmers need to apply more potassium fertilizer [15] to ensure the maximum crop yields [16,17]. However, overfertilization can lead to nutrient loss, the imbalance of physical and chemical properties, and the weakening of the biological properties of the soil [18], and it can be harmful to the environment [2,19]. Organic farming is gaining interest [20]; plant growth-promoting rhizobacteria (PGPR) are rhizosphere bacteria that promote plant growth and inhibit plant pathogens [21]. PGPR can meet the demand for “less fertilizer and more food”, which can reduce the use of fertilizer, thereby increasing soil fertility to ensure a crop yield [22].

*B. subtilis*, a common beneficial strain in plants [23], which is widely used [24]. *B. subtilis* can directly or indirectly affect the growth and development of plants, thus improving the crop yield [25,26], including enhanced nutrient absorption, plant hormone synthesis [27,28], iron carrier (iron fixation) production [29], soil nutrients (nitrogen, phosphorus, and potassium) solubilization [30,31,32], etc. Therefore, *B. subtilis* is often used in biological control agents and biofertilizers. Studies have shown that RB14, isolated from tomato roots by Zohora and others [33], is able to induce germination in tomatoes, resulting in a germination rate of more than 99%. Zhang [34] found that *B. subtilis* combined with two other rhizosphere growth-promoting bacteria inhibited soil-borne diseases of sweet pepper (*Capsicum frutescens* L.) and improved the soil chemistry, as well as fruit quality and soil properties. Jordan Vacheron et al. [35] found that *B. subtilis* could regulate the root system architecture, thereby promoting plant yield growth, when studying the interaction between plant growth-promoting bacteria in soil and various plants. Previous studies on *B. subtilis* mainly focused on screening efficient strains and the effects of using *B. subtilis* formulations alone on crops and soil, while there were few reports on the effects of *B. subtilis* combined with other fertilizers on plants and soil. Based on the above research status, this study took cucumber as the research object to study the effects of potassium combined with *B. subtilis* on cucumber growth and photosynthesis, in order to screen out the optimal fertilizer combination, and provide data support for the yield of cucumber cultivated in facilities, the development of an efficient microbial fertilizer, and the reduction of fertilizer use.

## 2. Materials and Methods

### 2.1. Preparation of B. subtilis Solution

The test strain was *Bacillus subtilis* KCKB1, which was stored in China General Microbiological Culture Collection Center (CGMCC; No. 1.3358). *B. subtilis* was striated on LB solid medium, activated 3 times in a constant temperature incubator at 26.5 °C to improve the activity of the strains; single colonies were selected with inoculating needle, transferred to 5 mL LB liquid medium, and shaken and cultured for 24 h in a constant temperature shaker at 26.5 °C and 166 r/min. A small amount of activated bacterial solution was added to 50 mL LB liquid medium and continued to be oscillated for 24 h under this condition, and a spectrophotometer was used to adjust the concentration of the bacterial solution to OD_600_ of 0.6–0.8 (Figure 1a).

### 2.2. Cultivation of Cucumber Seedlings

The seed of cucumber was Xinjin No. 4, which was derived from Jilin Provincial Key Laboratory of Plant Resources Science and Green Production, Jilin Normal University. First, rinsed cucumber seeds with tap water, soak in 75% alcohol by volume for 30 s, sterilize with 5% sodium hypochlorite by volume for 12 min, and then rinse in sterile water and soaked for 12 h, and, finally, place in an incubator at 28 °C. Then, 12 h later, the germinated cucumber seeds were planted in a seedling pot (28 × 25 cm). After 5 days of cucumber planting, cucumber seedlings with relatively stable growth status were selected, and 20 mL of bacterial solution was injected into the root every week, and the index was determined 5 weeks later.

### 2.3. Experimental Design

The experiment adopts a dual factor randomized block design. The two factors included different potassium supply levels and *B. subtilis*. Eight treatments were designed with six replicates per treatment (Figure 1b). The experimental treatment started at the four-leaf single-phase stage, and potassium sulfate containing 50% K_2_O was selected as potassium fertilizer. The following table shows the packet processing (Table 1).

### 2.4. Measurement of Growth Indicators

Plant height, leaf width, and leaf length were measured with a straightedge, wherein leaf area (cm^2^) = leaf length × leaf width × 0.75. Stem thickness of the plant was measured with a vernier caliper. The above-ground plants and roots were rinsed with water. After the fresh samples were degreened in the oven at 105 °C for 30 min, they were subsequently dried to constant weight at 70 °C. Finally, the dry weight of the roots and stems was measured using an electronic balance with an accuracy of 1/10,000. The roots of cucumber were scanned using Epson 11000 scanner (Epson America, Inc., Los Alamitos, CA, USA), and we analyzed the images by Win RHIZO 2012b root analysis system (Regent, Vancouver, BC, Canada), to obtain root morphology data such as root volume, root length, number of forks, etc.

### 2.5. Determination of K_2_O Content in Plants

Accurately weigh the mill and sift plant sample 0.05 g into a triangular flask; add a small amount of water to moisten; respectively, add 2.0 mL concentrated sulfuric acid and 10 drops of plant digestion accelerator; shake well; place a curved neck funnel at the mouth of the bottle; heat at a low temperature on an electric furnace; digest until the liquid in the bottle is clarified; remove and cool; fix the volume to 100 mL; shake well; and prepare the tested liquid. Then, 4 drops of plant nitrogen 1, 2, and 3 reagents were added in turn. After being shaken, they were left for 5 min, and then colorimetric analysis was carried out with a high-precision soil fertilizer nutrient detector (TY-04 +, Zhengzhou Tengyu Instrument Co., LTD., Zhengzhou, China) to measure the content of K_2_O in the plants. Each treatment was repeated six times.

### 2.6. Determination of Chlorophyll Content

Next, 1.0 g leaves of each sample were extracted in 10 mL 95% ethanol. The extracts were centrifuged at 10,000 rpm 25 °C for 10 min and the supernatant was retained. Measurement of absorbance at 663 nm and 645 nm was carried out using a spectrophotometer [36]. The unit of chlorophyll content was mg·g^−1^.
$$\mathrm{Total}\;\mathrm{chlorophyll}\;p\;(\mathrm{mg}\cdot L{-}^{1})\;=\;20.29A_{645}\;+\;8.05A_{663}\;\mathrm{Chlorophyll}\;\mathrm{content}\;(\mathrm{mg}\cdot g{-}^{1})=\frac{p\times V}{m\times 1000}$$
where: *p*: mass concentration of chlorophyll (mg·L^−1^); V: volume of sample extract solution (mL); *m*: sample mass (g); A_645_: absorbance at 645 nm; and A_663_: absorbance at 663 nm [36].

### 2.7. Determination of Gas Exchange Parameters

The leaf photosynthetic parameters were measured using CIRAS-3 Portable Photosynthesis Assay System (PP Systems, Amesbury, MA, USA), leaf chamber temperature was 25 °C, CO_2_ concentration was 400 μmol·mol^−1^, and flow rate was 500 μmol·s^−1^. Leaf gas exchange measurements were taken from the second and third leaves of each potted plant starting from the tip of the plant, and were carried out from 8 a.m. to 12 a.m. on sunny day [37]. The main parameters were net photosynthetic rate (A, μmol·m^−2^·s^−1^), transpiration rate (E, mmol·m^−2^·s^−1^), intercellular carbon dioxide concentration (C_i_, μmol·mol^−1^), stomatal conductivity (g_s_, mmol·m^−2^·s^−1^), and water use efficiency WUE (%). Biological replicates were performed 6 times per treatment.

### 2.8. Determination of Chlorophyll Fluorescence Parameters

The chlorophyll fluorescence parameters were determined by IMAGING-PAM modulated chlorophyll fluorescence imaging system (MINI-IMAGING-PAM, Heinz Walz, Effeltrich, Germany) [38]. The leaves were dark-adapted for 30 min before measurement, and the maximum quantum yield, non-photochemical quenching coefficient, actual quantum yield, and electron transfer rate of photosystem II (PSII) were measured, respectively. The parameters and calculation formulae are shown in Table 2. To determine the effect of irradiation level on PSII fluorescence emission, a 16-step fast light curve (rlc) was performed with each photosynthetically active radiation (PAR) level lasting 20 s at (0, 20, 55, 110, 185, 280, 335, 395, 460, 530, 610, 700, 800, 925, 1075, and 1250 μmol photons·m^−2^·s^−1^), respectively.

### 2.9. Data Analysis and Statistics

IBM SPSS 22.0 was used for statistical analysis, and linear model two-factor analysis of variance was used to analyze the effects of *B. subtilis* and the amount of potassium fertilizer applied. At the same time, Duncan method (one-way ANOVA) was used to analyze the significance of the difference between *B. subtilis* and no *B. subtilis* applied and the difference between different potassium supply levels (*p* ≤ 0.05). The chart was produced by OriginPro 2022 (OriginLab, Northampton, MA, USA).

## 3. Results

### 3.1. Growth Index

The above-ground dry weight, plant height and stem thickness are important indices of cucumber seedling growth and development. In the pot experiment, the growth of cucumber seedlings was measured under different potassium supply levels combined with *B. subtilis*. The results showed (Figure 2) that potassium, *B. subtilis*, and their combined influence significantly impacted various plant growth parameters including the plant height, leaf area, and above-ground and root dry weight. Potassium and *B. subtilis* had a significant effect on the stem diameter of cucumber seedlings; however, their interaction did not yield any significant impact on the stem diameter. Under different potassium supply levels, there were significant differences between treatments with *B. subtilis* application (TK) (*p* ≤ 0.05); the highest plant height was recorded in the K2 treatment, which was significantly higher than that of the K0, K1, and K3 treatments, increasing by 101.78%, 28.5%, and 49.64%, respectively. In the control treatment (CK), the plant height was significantly greater in the K2 treatment compared to the K0, K1, and K3 treatments (*p* ≤ 0.05), which was 108.58%, 56.44%, and 69.96% higher, respectively (Figure 2a). Under the TK treatment, the leaf area of the K1, K2, and K3 treatments was significantly larger than that of the K0 treatment (*p* ≤ 0.05), increasing by 59.72%, 116.32%, and 78.78%, respectively, and reached the maximum under the K2 treatment, which was 113.32 cm^2^. Under the CK treatment, all treatments exhibited a significant difference in leaf area (*p* ≤ 0.05), with the K2 treatment showing a considerably bigger difference than the K0, K1, and K3 treatments (*p* ≤ 0.05), increasing by 158.47%, 64.26%, and 42.42%, respectively (Figure 2c). Potassium application significantly increased the stem diameter, and above-ground part and root dry weight at the seedling stage of the cucumber under all treatments (*p* ≤ 0.05), but there was no obvious distinction between the K1 and K3 treatments. The cucumber under the B2 treatment had the best growth status, and stem diameter, above-ground part, and root dry weight of cucumber were 6.42 cm, 1.37 g, and 0.16 g, respectively (Figure 2b,d,e). With the exception of the stem diameter during the K0 treatment, there was no discernible difference between the TK and CK treatments; in the TK group, the cucumber seedlings exhibited a substantially higher plant height, stem diameter, leaf area, and above-ground part and root dry weight compared to the CK group (*p* < 0.05).

### 3.2. Root Morphology

Plant roots have the ability to absorb water and nutrients from the soil, allowing plants to become firmly established in it. Their morphological characteristics are intimately linked to how plants grow and develop, and the root morphology index can reflect the growth of plant roots well. As can be seen from Table 3, there were notable differences in total root length, root volume, root surface area, branch number, and cross number between *B. subtilis* and potassium (*p* < 0.01); however, their interactions had no appreciable effects on the cucumber’s root volume, root surface area, branch number, and cross number. In cucumber seedlings treated with K1, the total root length, root surface area, root volume, cross number, and branch number increased significantly by 98.08%, 94.95%, 133.63%, 74.73%, and 123.81%, respectively, when compared to K0 (*p* ≤ 0.05). Under the K2 treatment, the total root length, root surface area, root volume, cross number and branch number of cucumber seedlings were significantly increased by 196.69%, 210.33%, 277.01%, 210.7%, and 284.39% compared with the K0 treatment, respectively (*p* ≤ 0.05). Under the K3 treatment, the total root length, root surface area and cross number of cucumber seedlings were significantly increased by 65.4%, 46.22%, and 112.39% compared with the K0 treatment (*p* ≤ 0.05), and the root volume and cross number of cucumber seedlings had no significant difference, but were improved. When *B. subtilis* was applied, cucumber seedlings’ total root length, root surface area, and cross number were noticeably greater than when no treatment was applied. The root morphological indices of the cucumber seedlings treated with B2 had the highest values among them, which were 109.58 cm, 149.2 cm^2^, 1.61 cm^3^, 8036.83, and 1759.67. In conclusion, K had a significant growth-promoting effect on increasing the total root length, total root surface area, total root volume, and cross number of cucumber seedlings. Cucumber seedling roots grew more quickly after inoculation, which had a beneficial synergistic effect.

### 3.3. Photosynthetic Characteristics

The intercellular CO_2_ concentration, net photosynthetic rate, transpiration rate, water use efficiency, and stomatal conductance on cucumber seedlings were all significantly influenced by potassium, while *B. subtilis* significantly affected all indices except water use efficiency, and the two factors showed extremely significant interactions with the photosynthetic characteristics of cucumber seedlings (Figure 3). These results indicated that potassium and *B. subtilis* could promote the photosynthesis of cucumber seedlings to a certain extent. The transpiration rate refers to the amount of water lost by plants per unit time, expressed by a weight ratio. Potassium application reduced cucumber seedling E. The order of the CK treatment’s transpiration rate was K3 > K1 > K0 > K2, and the overall trend was first rising, then decreasing, and then rising. After the TK treatment, cucumber seedlings’ transpiration rate exhibited a trend of first declining and then rising. When compared to the K1, K2, and K3 treatment groups, the E of the K0 treatment increased by 21.86%, 45.00%, and 6.07%, respectively, and was considerably greater than that of all fertilization treatments (*p* ≤ 0.05) (Figure 3b). Under the TK treatment, Ci rose as the K application amount increased and started to decrease at a specific concentration. The K1, K2, and K3 treatment groups increased by 19.20%, 19.60%, and 16.13% compared with K0, respectively, and there were noticeable differences (*p* ≤ 0.05). Additionally, there were some variations in the CK treatments; the K1 and K2 treatment groups were significantly higher than K0. The increases were 17.64% and 23.82% (*p* ≤ 0.05) (Figure 3d); the data in the figure showed that stomatal conductance increased after potassium application. When the potassium application rate was 0.2 g/pot and *B. subtilis* was applied at the same time, g_s_ reached the maximum value of 117.00 m/s, while the continuous increase in the potassium application rate decreased—g_s_. The variation trends of the water use efficiency and net photosynthetic rate among different treatments were consistent with the transpiration rate (Figure 3a,c,e).

### 3.4. Chlorophyll a Fluorescence Parameter

In the TK treatment, the order of the Fo and Y (NO) of cucumber seedlings was K3 > K0 > K2 > K1; compared to other treatments, the B3 treatment was noticeably higher (*p* ≤ 0.05), and the overall trend was first decreased and then increased. In the CK treatment, Fo was substantially greater than that of the other three groups following the K2 treatment, Y (NO) was substantially greater in the K1 group than in the other three treatments (*p* ≤ 0.05); the Fo and Y (NO) of cucumber seedlings treated with K0, K1, and K2 CK were notably greater than those TK treatments, while K3 demonstrated that the TK treatment was considerably more than the CK treatment (Figure 4a,f). In the TK treatment, the Fv/Fm and Fv/Fo of cucumber seedlings under different K levels were significantly different (*p* ≤ 0.05), with K1 > K0 > K2 > K3, and the general trend went from increasing to decreasing. In the CK treatment, the cucumber seedlings’ Fv/Fm and Fv/Fo were in the order of K3 > K0 > K1 > K2. Compared to other treatments, the K2 treatment was noticeably lower (*p* ≤ 0.05). The overall trend was first decreased and then increased. Cucumber seedlings’ Fv/Fm and Fv/Fo treated with K0, K1, and K2 TK were notably greater than those CK treatments, while K3 demonstrated that the CK treatment was considerably more than the TK treatment (Figure 4c,g).

In the TK treatment, the chlorophyll content of cucumber seedlings was ranked as K2 > K1 > K0 > K3, and, compared to other treatments, that of the B2 treatment was noticeably higher (*p* ≤ 0.05), exhibiting an increasing and subsequently declining trend. Under different potassium levels, the chlorophyll content of cucumber seedlings was in the order of K2 > K1 > K0 > K3, and cucumber seedlings treated with K2 had a considerably higher chlorophyll content than seedlings treated with other treatments (*p* ≤ 0.05), and demonstrated an increasing and subsequently declining trend. The experimental group’s and the control group’s levels of chlorophyll varied significantly (*p* ≤ 0.05). And, under the TK treatment, there was a significant increase in the chlorophyll content compared to the CK treatment (Figure 4h).

In the TK treatment, PSII and Fv′/Fm′ were K0 > K1 > K2 > K3, and, as the amount of potassium applied increased, the chlorophyll content decreased. In the CK treatment, between various potassium application amounts, there were notable variations in the Fm and Fv′/Fm′ (*p* ≤ 0.05). The Fm and Fv′/Fm′ of the K0 and K1 TK treatments were notably greater than the results of the CK treatments, and the K2 and K3 CK treatments had a greater Fm and ΦPSII than the TK treatments. The ΦPSII of the K1 and K2 CK treatments were notably greater than the results of the TK treatments, while the Fv′/Fm′ was opposite to that in the K1 and K2 treatment (Figure 4b,d,e).

We also analyzed the photoresponse curves of ETR, NPQ, and qP of PSII under different potassium levels. The ETR curve was positively correlated with the photosynthetic rate. With the increase in light intensity, the ETR treated by CK increased faster than that treated by TK. Among them, the ETR photoresponse curve treated by K0 showed the best state, and the ETR treated by CK increased the most (Figure 4i). Under light intensity conditions (PAR < 300 μmol·m^−2^·s^−1^), the TK treatment’s NPQ was noticeably greater than the CK treatment’s (Figure 4j). With the increase in light intensity, the qP of the TK treatment decreased less than that of the CK treatment (Figure 4k). In different treatments at different potassium levels, the excess excitation energy is dissipated as heat energy. Therefore, inoculation with *B. subtilis* can alleviate PSII damage to a certain extent.

Under different potassium levels, potassium moved from the cucumber’s root system to its above-ground tissue more readily than that under the treatment of *B. subtilis*, as shown in Figure 5. Under the condition of applying *B. subtilis* (TK), compared with the K0 and K3 treatments, the root potassium content after the K2 treatment was significantly increased by 63.52% and 79.18%, respectively, but, in contrast to the K1 treatment, the increase was not statistically significant. Under the control treatment (CK), the root total potassium content reached the maximum under the K1 treatment, which was significantly higher than that under the K0 and K3 treatment, increasing by 24.40% and 27.01%, respectively. There were significant differences in the above-ground total K content after the CK treatment, and it reached the maximum under the K2 treatment, which was significantly increased by 45.60%, 20.84%, and 50.17% compared with the K0, K1, and K3 treatments. Under the TK treatment, there was no discernible difference among all treatments, but, under the K2 treatment, the highest potassium content was 295.5 mg/kg. Compared with the CK treatment, the total K content of plants in the TK group increased significantly after each treatment, and the increase was K1 > K0 > K2, while the total K content of plants under the K3 treatment decreased, indicating that the application of *B. subtilis* could encourage cucumber seedlings to absorb K.

### 3.5. Influencing Factors of Leaf Growth Index and Photosynthetic Parameters of Cucumber

A principal component analysis and correlation analysis were conducted on 23 indices of cucumber seedlings, including the plant height, stem diameter, above-ground and root dry weight, Ci, gs, A, WUE, total root length, root surface area, root volume, chlorophyll content, Fm, Fo, etc. The results were displayed in Figure 6 and Figure 7. As the indices of each treatment are different, it is one-sided to use a certain index to evaluate the best inoculation treatment of cucumbers. Therefore, through the principal component analysis (Figure 6), it was found that the first and second ranking axes explained 61.8% and 16.5% of the variance variation of the total variability, respectively, and the first four axes’ combined explanation percentage was 78.3%, indicating a good ranking effect. From left to right along axis 1, the E, Y(NO), Fv/Fm, and Fv/Fo decreased, while the Fv′/Fm′, Fm, Fo, plant height, stem diameter, dry weight of above ground, Ci, g_s_, A, WUE, and root morphological indices increased.

Correlation analysis showed that (Figure 7) the plant height, stem diameter, leaf area, and above-ground and root dry weight were significantly positively correlated with the Ci, g_s_, A, WUE, total root length, root surface area, root volume, branch number, crossover number, chlorophyll content, Fm, and Fo (*p* ≤ 0.01). The root morphological indices were significantly positively correlated with the above-ground morphological indices, Ci, g_s_, A, WUE, and chlorophyll content of cucumber seedlings (*p* ≤ 0.05). The E was positively correlated with the Fv/Fm, Y(NO), Fv/Fo, Fv′/Fm′, and negatively correlated with 18 indices such as the plant height, stem diameter, total root length, and WUE. The Y(NO), Fv/Fo, and Fv/Fm were negatively correlated with the root morphological indices, growth indices, chlorophyll content, Ci, g_s_, A, WUE, Fm, and Fo. The Fv′/Fm′ exhibited a negative correlation with the Y(II) and Fo, a positive correlation with the Fv/Fo and Fv/Fm, and a weak correlation with the remaining 17 indices.

## 4. Discussion

In recent years, people are increasingly interested in PGPR. Using PGPR to solubilize slow and ineffective potassium in soil, it provides an effective way for plants to assimilate and exploit nutrients. Simultaneously, PGPR has the capability to enhance plant growth and development. In the realm of agricultural production, PGPR can reduce the use of chemical fertilizer and gradually become a substitute for chemical fertilizer [30,32]. Therefore, this research evaluated the solubilization effect of *B. subtilis* on potassium and its promotion effect on cucumber seedling growth.

The research indicates that applying PGPR can encourage crops to take up potassium from the soil, which will, in turn, encourage crop growth [3,39]. The results demonstrated that the growth indices of plants injected with *B. subtilis* under various potassium levels were considerably greater compared with those of plants treated without *B. subtilis*. (Figure 2). The application of *B. subtilis* and the selection of the potassium level both significantly impacted the dry weight. This result is in line with earlier research by Blake et al., which discovered that *B. subtilis* secretes hormones like IAA that stimulate plant growth [23,40]. An essential organ for the growth and development of plants is the root system. A large root system can better hold the plant body and promote the absorption of water and nutrients from the soil by the plant. The quality of plant growth can be directly reflected from the root morphological changes [41]. After being inoculated with *B. subtilis*, studies have revealed that *Arabidopsis thaliana* has significantly increased the dry weight and root morphological indices [42]. Studies have shown that the application of *B. subtilis* microbial inoculants can increase the soil nutrient content, thus promoting the growth of tomato. *B. subtilis* was applied to improve plant stress resistance [43,44]. Our study revealed that the morphological parameters of roots treated with *B. subtilis* were superior to those treated without *B. subtilis* inoculation (Table 3). As the energy source and physiological foundation of plants, photosynthesis supplies nutrients necessary for the growth and development of plants [45]. The amount of chlorophyll is a crucial indicator with which to judge the normal physiological metabolism and photosynthesis of plants [46], because it plays an indispensable role in the process of converting solar energy into organic matter [47]. Singh found that *B. subtilis* CIM considerably raised the amount of chlorophyll in rice leaves [45]. The results obtained were consistent with those of Zhang et al. [48]. Zhang et al. increased the total chlorophyll and carotenoid contents of *Arabidopsis thaliana* by applying *B. subtilis GB03*, a soil bacterium that promotes plant growth, and significantly improved the photosynthetic capacity. Among various physiological index parameters, the chlorophyll content can directly affect the photosynthetic efficiency of plants [49]. Our study found that, compared with the untreated treatment, the Ci, g_s_, and WUE of cucumber seedlings treated by *B. subtilis* were higher, which proved that more carbon dioxide involved in photosynthesis was removed, and the water loss by transpiration was reduced, which agreed with Bolton and Bilgin’s research findings [50,51]. While most of the sunlight is emitted as heat dissipation or chlorophyll fluorescence, a tiny amount is absorbed by chlorophyll molecules and initiates photochemical reactions during electron transport during photosynthesis [52]. Chlorophyll fluorescence is an indicator of photochemical efficiency, representing photosynthetic capacity and energy conversion efficiency, which can be determined by measuring the Fm, Fo, Y(NO), Fv/Fm, Fv/Fo, Fv′/Fm′, ΦPSII, NPQ, and ETR [53]. Some studies have shown that, under drought stress, the inoculation of *B. subtilis* can mitigate the detrimental effects of drought on parameters related to chlorophyll fluorescence [52]. We found that, under different potassium levels, cucumber seedlings inoculated with *B. subtilis* showed considerably greater values of Fm, Fv/Fm, Fv/Fo, and Fv′/Fm′ compared to the untreated treatment, while the values of Fo, Y(NO), and ΦPSII were lower than those in the untreated treatment (Figure 3 and Figure 4).

Through a principal component analysis, 23 growth indicators of eight treated cucumber seedlings were converted into two principal components, and the principal component load matrix reflected the relative strength and direction of action of each index on the principal component load, that is, the degree of influence of the index on the principal component. The cumulative contribution rate of these two principal components was 78.3%, which could represent 78.3% of the total index information. According to the principal component analysis, he cucumber seedling height, stem diameter, leaf area, and above-ground dry weight significantly affected other parameters, while the root volume, net photosynthetic rate, and chlorophyll content could, in turn, promote the overall growth and development of the plant (Figure 6).

## 5. Conclusions

Potassium ion is the main element for crop growth. This experiment studied the potassium and *B. subtilis*’ effects on cucumber seedling growth and photosynthesis. The results showed that the application of 0.2 g/pot potassium and *B. subtilis* significantly improved the above-ground morphological indices, root morphological indices, and photosynthetic parameters of cucumber. The results of this study indicated that *B. subtilis* significantly promoted the growth of cucumber seedlings, and provided a theoretical basis for further utilizing *B. subtilis* to make microbial fertilizer and improve the nutrient absorption efficiency of cucumber. This can be used to promote the development of agriculture.

## Figures and Tables

**Figure 1 biology-13-00348-f001:**
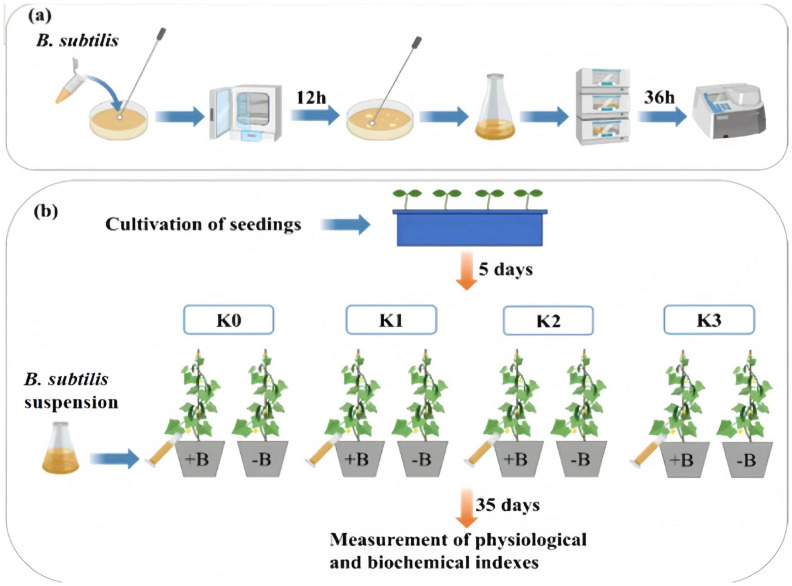
Flow chart of experimental design. +B, inoculate *B. subtilis*; −B, Not vaccinated with *B. subtilis*. (**a**) Preparation of *B. subtilis* suspension; (**b**) Pot experiment.

**Figure 2 biology-13-00348-f002:**
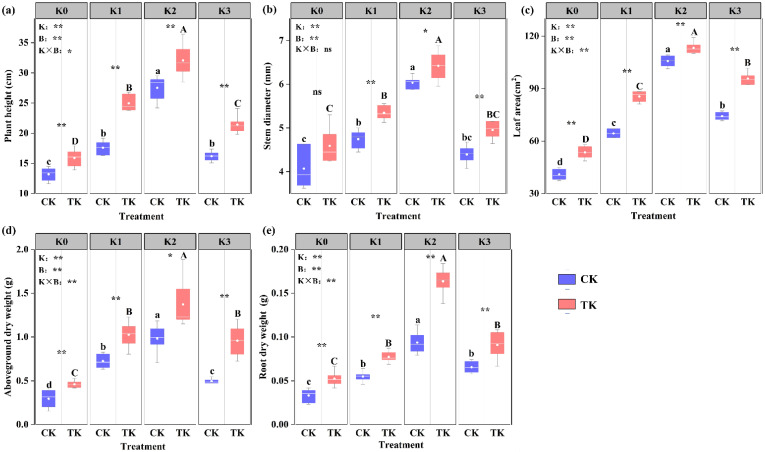
Effects of potassium fertilizer combined with *B. subtilis* on cucumber seedling growth: (**a**), plant height; (**b**), stem diameter; (**c**), leaf area; (**d**), above-ground dry weight; and (**e**), root dry weight. Different capital letters indicated the significant difference under the treatment of *B. subtilis* (B0, B1, B2, B3) (*p* ≤ 0.05); The significant difference (*p* ≤ 0.05) under various potassium fertilizer dosages was indicated by different lowercase letters (no potassium K0, 0.1 g potassium K1, 0.2 g potassium K2 and 0.3 g potassium K3); *, ** indicated that the control group (CK) and the experimental group (TK) differed significantly at *p* = 0.05 and *p* = 0.01 in turn, while ns indicated no significant differences (*p* > 0.05). K, B and K × B respectively indicated the significance of difference in F value of Duncan test of the effects of potassium level, *B. subtilis*, potassium level × *B. subtilis* on cucumber morphological indices, the same as below.

**Figure 3 biology-13-00348-f003:**
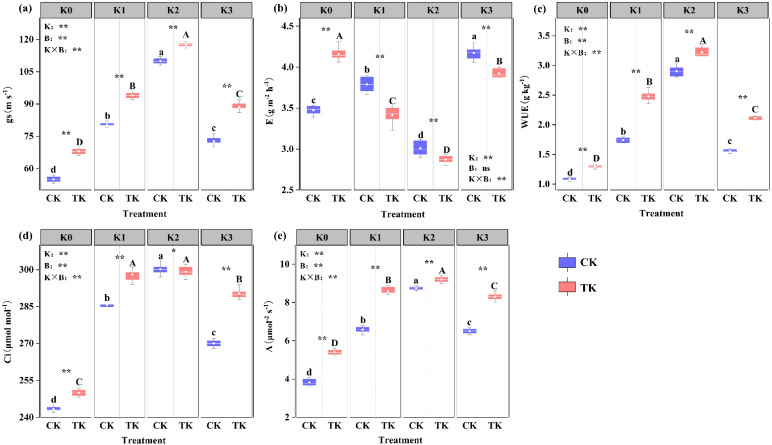
Effects of potassium fertilizer combined with *B. subtilis* on photosynthetic characteristics of cucumber seedlings: (**a**), stomatal conductivity; (**b**), transpiration rate; (**c**), water use efficiency; (**d**), intercellular CO_2_ concentration; and (**e**), net photosynthetic rate. (a–d), different letters indicate a significant difference between treatments without applying *B. subtilis*; (A–D), Different letters indicate significant differences between treatments in which *B. subtilis* is applied; * *p* ≤ 0.05; ** *p* ≤ 0.01.

**Figure 4 biology-13-00348-f004:**
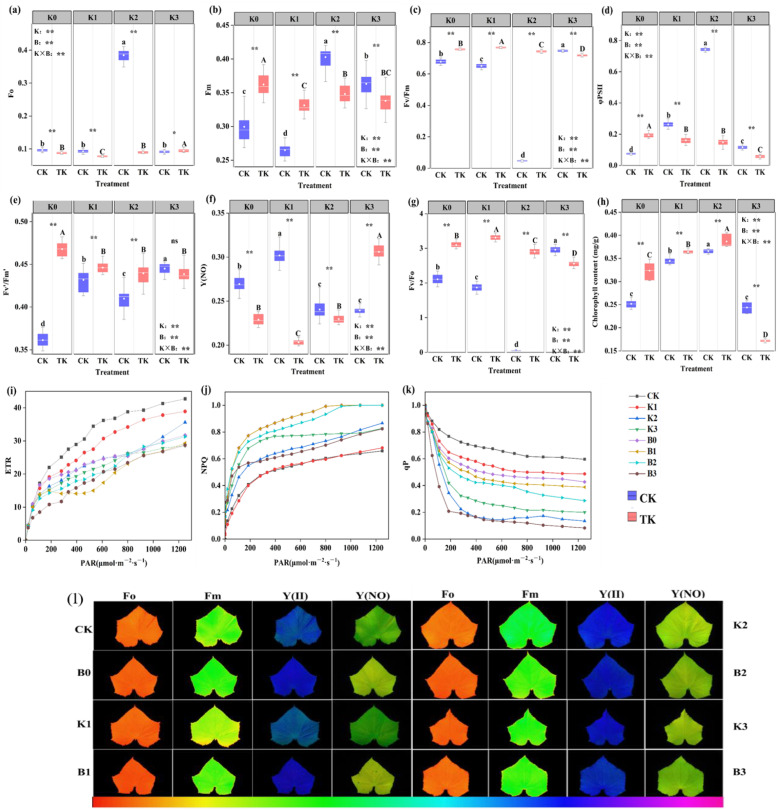
Effects of potassium fertilizer combined with *B. subtilis* on chlorophyll a fluorescence parameters of cucumber seedlings: (**a**), initial fluorescence; (**b**), maximum fluorescence; (**c**), maximum quantum yield of PSII; (**d**), actual quantum yield of PSII; (**e**), effective quantum yield of PSII; (**f**), non-regulatory energy dissipation; (**g**), potential activity of PSII; (**h**), chlorophyll content; (**i**), photosynthetic electron transport rate; (**j**), non-photochemical quenching coefficient; and (**k**), photochemical quenching coefficient; (**l**), Chlorophyll fluorescence image; (a–d), different letters indicate a significant difference between treatments without applying *B. subtilis*; (A–D), Different letters indicate significant differences between treatments in which *B. subtilis* is applied; * *p* ≤ 0.05; ** *p* ≤ 0.01.

**Figure 5 biology-13-00348-f005:**
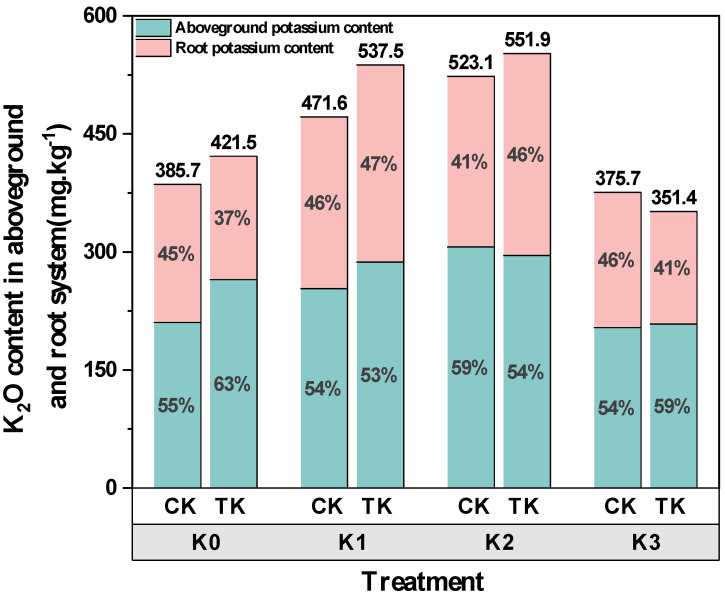
K_2_O content in the ground part and root of cucumber seedlings.

**Figure 6 biology-13-00348-f006:**
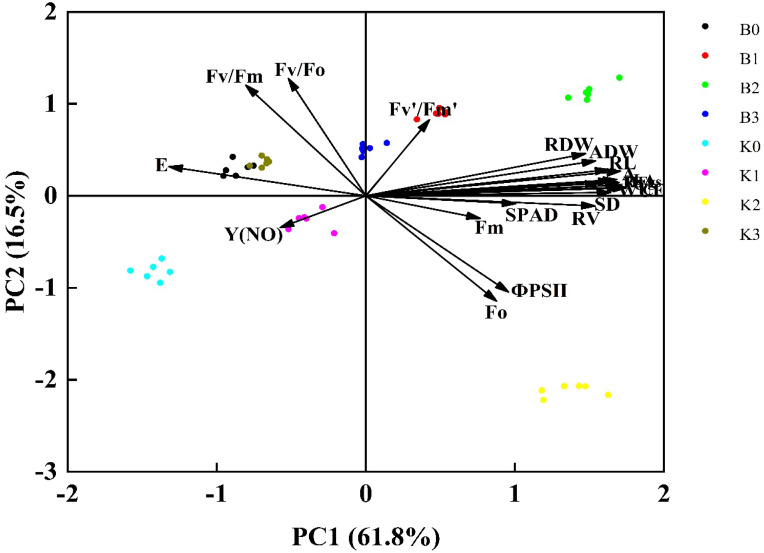
Principal component analysis of each index.

**Figure 7 biology-13-00348-f007:**
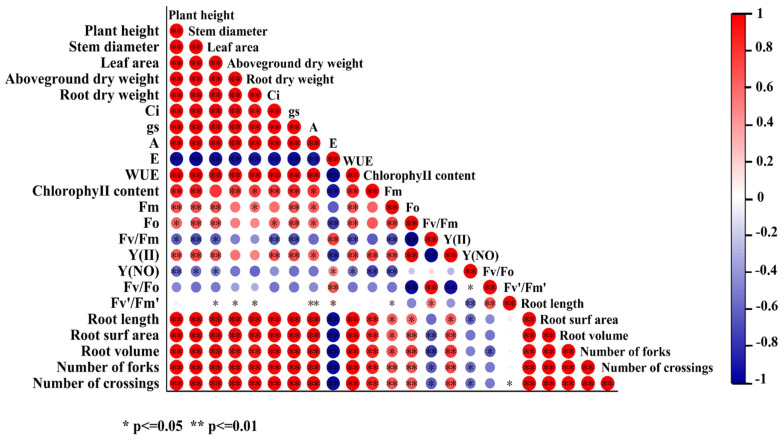
Correlation analysis of physiological and biochemical indices.

**Table 1 biology-13-00348-t001:** Grouping setting of pot experiment.

Treatment	Potassium Application Rate (g/pot)	*B. subtilis* Suspension (1.5 × 10^6^ CFU/mL)
K0CK	0	0
K0TK	0	20 mL
K1CK	0.1	0
K1TK	0.1	20 mL
K2CK	0.2	0
K2TK	0.2	20 mL
K3CK	0.3	0
K2TK	0.3	20 mL

**Table 2 biology-13-00348-t002:** Chlorophyll fluorescence parameters and calculation equations.

Formula	Parameters
Fo	Minimum fluorescence
Fm	Maximum fluorescence
Fm′	Maximum fluorescence under acting light
Fo′	Minimum fluorescence under acting light
Fs	Steady-state fluorescence level
Y(NO)	Non-regulatory energy dissipation
Fv = Fm − Fo	Maximum variable fluorescence
Fv′ = Fm′ − Fo′	Maximum variable fluorescence intensity under acting light
Fv/Fm = (Fm − Fo)/Fm	Maximum quantum yield of PSII
Fv′/Fm′ = (Fm′ − Fo′)/Fm′	Effective quantum yield of PSII
ΦPSII = (Fm′ − Fs)/Fm′	Actual quantum yield of PSII
qP = (Fm′ − Fs)/(Fm′ − Fo′)	Photochemical quenching coefficient
Fv/Fo = (Fm − Fo)/Fo	Potential activity
NPQ = (Fm − Fm′)/Fm′	Non-photochemical quenching coefficient
ETR = PAR × ΦPSII × 0.84 × 0.5	Non-photochemical quenching coefficient

**Table 3 biology-13-00348-t003:** Effects of different potassium levels combined with *B. subtilis* on morphological indicators of cucumber root system.

Treatment	Root Length (cm)	Root Surf Area (cm^2^)	Root Volume (cm^3^)	Number of Forks	Number of Crossings
K0CK	283.28 ± 6.76 h	43.37 ± 2.17 e	0.46 ± 0.09 f	2286.17 ± 73.82 f	414.33 ± 19.62 f
K0TK	398.22 ± 10.63 g	64.30 ± 4.63 d	0.83 ± 0.12 def	3146.67 ± 262.81 ef	650 ± 37.90 e
K1CK	561.13 ± 10.02 e	84.56 ± 6.56 c	1.08 ± 0.14 cde	3994.67 ± 280.55 e	927.33 ± 35.74 d
K1TK	755.66 ± 9.63 c	110.29 ± 2.82 b	1.25 ± 0.09 bc	6100.83 ± 102.05 c	1376.67 ± 31.39 b
K2CK	841.14 ± 14.83 b	134.60 ± 8.73 a	1.74 ± 0.21 a	7103.17 ± 529.18 b	1592.67 ± 132.98 a
K2TK	1098.58 ± 31.25 a	149.20 ± 4.14 a	1.61 ± 0.06 ab	8036.83 ± 149.37 a	1759.67 ± 21.70 a
K3CK	468.55 ± 8.72 f	63.42 ± 1.73 d	0.71 ± 0.05 ef	3400.83 ± 177.54 ef	880 ± 11.97 d
K3TK	653.16 ± 9.06 d	99.17 ± 1.61 bc	1.16 ± 0.04 cd	4932.83 ± 40.19 d	1138.5 ± 37.9 c
KCK	***	***	***	***	***
KTK	***	***	**	***	***
KCK × KTK	***	ns	ns	ns	ns

Note: (a–h), different letters denote significant differences between treatments; **, *p* ≤ 0.01; ***, *p* ≤ 0.001; ns: not significant.

## Data Availability

The data used in this study are available from the corresponding author upon the submission of a reasonable request.

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
