# Peer review of "Effects of Bacillus subtilis on Cucumber Seedling Growth and Photosynthetic System under Different Potassium Ion Levels"

_biology, 2024, doi:10.3390/biology13050348_

Round 1
Reviewer 1 Report
Comments and Suggestions for Authors
The peer-reviewed article is experimental, dedicated to the study of the influence of different concentrations of potassium ion together with the culture of Bacillus subtilis on the growth indicators of cucumbers. A relevant and important topic is considered, but there are significant remarks to the work:
1) Almost everywhere in the text of the article, the influence of potassium, not potassium ions, is indicated, and this is a significant error, since not a elemental potassium was used as a fertilizer, but a potassium salt. Therefore, it is necessary to write potassium ions, including in the title of the article. In addition, the authors note that the fertilizer contained KO2, and it is known from the literature that KO2 also exhibits the properties of a fertilizer. However, such studies are not discussed in the reviewed work and attention should be paid to this. In the article, the authors should indicate not the mass of fertilizer per gram of soil, but the concentration of potassium ions and KO2 for each variant of the study.
2) Lines 81-82. The full name of the tested strain of Bacillus subtilis must be specified. However, if it is Bacillus subtilis subsp. subtilis NCIB 3610 (CGMCC, No. 1.3358), which logically follows from the information provided in the article, justification for the use of this particular strain should be provided, since pathogenic properties have been noted for it (doi: 10.3389/fcimb.2019.00183). At the same time, microorganisms used as biofertilizers should not pose a risk to animal and human health (doi: 10.1007/s12298-022-01138-y). This should be indicated in the Introduction and Discussion.
3) Lines 298-299. – Therefore, inoculation with Bacillus subtilis can alleviate PSII damage to a certain extent. - That is, the use of Bacillus subtilis can impair photosynthetic activity? Please explain, because the abstract notes the positive effect of these bacteria on leaf photosynthesis.
4) The conclusion should summarize the results of own research, provide recommendations regarding the prospects for further research. It is not clear why the conclusion indicates a decrease in the consumption of potash (perhaps better potassium salts?) by B. subtilis, because such studies were not conducted in the peer-reviewed work.
Other comments are technical and relate to the design of the article.
The Latin name of the bacteria is Bacillus subtilis in full only at the first mention, and then it is abbreviated as B. subtilis.
Lines 42. Potassium is a large number of elements necessary to affect plant growth and metabolism. - Element Potassium is one, not many. Please correct the expression.
Lines 80-88, Fig. 1(b). Not Bacillus subtilis solution. You need to write Bacillus subtilis suspension.
Fig. 1(b). The explanation of the experimental design and Fig. 1(b), various notations. Please approve. +B, -B should be explained in the caption to the figure.
Line 104. СК, TK, B0, K0 - there are no such markings in this section, although there are in the text of the article. Should be corrected.
Lines 129-130. To determine the content of chlorophyll, a formula was used (indicate the reference from which this formula was taken): write the formula itself, and below it what the notations mean.
Lines 132-140. The reference of the methodology should be indicated.
Table 1. Conventional designations must be deciphered: Fm, Fo, Fm', Fo', Fs, Fv, Fv', *. Cyrillic letter is used in the designation ФPSâ…¡ (Table 1, Lines 377, 382 also), although φPSII is written in places (Line 286, 287). Please pay attention to this.
Lines 236, 240, 242, 249, 258, 310, 317, 318. Potassium application reduced cucumber seedling E (Line 236). ..., the E of the K0... (Line 240). Under TK treatment, Ci rose... (Line 242). ...time, gs reached... (Line 249). What are the symbols E, Ci, gs F0, Y(NO), A, WUE, Pii, SPAD? In the Materials and methods section, it is necessary to indicate all the indicators that were studied and their interpretation.
Lines 326, 330, 384. 18 indices, 17 indices, 23 growth indicators? In the Materials and methods section, it is necessary to indicate all the indicators that were studied and their interpretation.
Table 2. An explanation should be provided in the notes to this table for the designations: **, ***, ns.
The peer-reviewed article can be published only after correcting all the noted comments.
Reviewer 2 Report
Comments and Suggestions for Authors
Highlights and overall thoughts:
· The manuscript presents the importance of the use of microorganisms to improve the use of fertilizer in crops.
· There are grammatical errors through the manuscript that can be solved with the review of an English speaker.
· In some sections, the order of the paragraphs or ideas does not follow a logical line of though.
Specific:
Abstract: Don’t use words as “more obvious” to describe the effect of your treatments; you can use other words like “evident”, “greater”, “significant”, etc.
Line 12: “to solubilize” instead of “to soluble”
Line 22: “p ≤ 0.05” instead of “P < 0.05” through all the manuscript
Lines 35-38: The 2021 production data is written in future tense, which make me think this information comes from years before 2021 and the data may be outdated. Use recent confirmed data (2023 production data).
Line 42-43: This sentence is confusing and needs to be rewritten for more clarity.
Lines 46-47: What do you mean by slow and relatively ineffective potassium? Please explain it in the text.
Line 53: The part of potassium sulfate being harmful needs citation.
Line 58: Bacillus subtilis is a Gram-positive beneficial bacterium, which…
Line 83: What do you mean by activated for 3 times? Three repetitions? Make it clear in the manuscript.
Materials and Methods: This section could be improved. At some points the sentences are hard to understand.
Line 207: Is there a reason why the p-value is different from the other analyses?
Table 2: In a footnote indicate what is the meaning of ns, the different letters, and ***
Discussion: You could discuss with other previous research done in cucumber or other similar crops. Arabidopsis thaliana is a great model plant, but there should be other more closely raleted examples.
Line 355, 365, 366: Arabidopsis thaliana and Bacillus subtilis must be in cursive
Revise References thoroughly
Comments on the Quality of English LanguageThere are some grammatical issues in the manuscript. I suggest revision.
Round 2
Reviewer 1 Report
Comments and Suggestions for Authors
Thank you for your article!
Line 42 - element, not elements
Table 2 - please swap either the names of the columns or their contents